ecology/health and disease and epidemiology/statistics

yellow fever, vulnerability, environment, cumulative logit model

**Author for correspondence:**
Joseph L. Servadio
e-mail: serva024@umn.edu

# Environmental determinants predicting population vulnerability to high yellow fever incidence

Joseph L. Servadio[1,2], Claudia Muñoz-Zanzi[1] and Matteo Convertino[3]

[1]Division of Environmental Health Sciences, School of Public Health, University of Minnesota, Minneapolis, MN, USA
[2]Center for Infectious Disease Dynamics and Department of Biology, Pennsylvania State University, University Park, PA, USA
[3]Future Ecosystems Lab, Tsinghua SIGS, Tsinghua University, Shenzhen, People's Republic of China

JLS, 0000-0002-9988-5712

Yellow fever (YF) is an endemic mosquito-borne disease in Brazil, though many locations have not observed cases in recent decades. Some locations with low disease burden may resemble locations with higher disease burden through environmental and ecohydrological characteristics, which are known to impact YF burden, motivating increased or continued prevention measures such as vaccination, mosquito control or surveillance. This study aimed to use environmental characteristics to estimate vulnerability to observing high YF burden among all Brazilian municipalities. Vulnerability was defined in three categories based on yearly incidence between 2000 and 2017: minimal, low and high vulnerability. A cumulative logit model was fit to these categories using environmental and ecohydrological predictors, selecting those that provided the most accurate model fit. Per cent of days with precipitation, mean temperature, biome, population density, elevation, vegetation and nearby disease occurrence were included in best-fitting models. Model results were applied to estimate vulnerability nationwide. Municipalities with highest probability of observing high vulnerability was found in the North and Central-West (2000–2016) as well as the Southeast (2017) regions. Results of this study serve to identify specific locations to prioritize new or ongoing surveillance and prevention of YF based on underlying ecohydrological conditions.

# 1. Introduction

Yellow fever (YF), a disease caused by yellow fever virus, is a prevalent public health concern in much of the tropics. It is endemic in sub-Saharan Africa and South America and spread by multiple genera of mosquitoes [1,2]. Infections can present as a mild, febrile illness, with a fraction of cases becoming severe and potentially fatal [1,3]. The existence of non-human primates as a reservoir and the ability for mosquitoes to transmit the virus transovarially [4,5] lead to the belief that the virus will never be globally eradicated [6]. Infections can, however, be reduced in humans due to the existence of a safe and effective vaccine [4,7].

Anticipated presence of YF in perpetuity motivates informed preparations to prevent future human cases, which can include vaccine prioritization, vector control, surveillance improvements or prioritized messaging. In order to most efficiently target preparations and conserve resources, identifying specific locations most vulnerable to high YF burden is of importance. This is of particular importance in locations where YF has been endemic or has seen recent epidemics, such as Brazil. In recent decades, YF has been endemic in Brazil, primarily occurring in the North and Central-West regions, adjacent to the Amazon. In 2016, a major epidemic began [8], where more cases were seen in a single year than had been seen in decades prior. The epidemic also largely occurred in the Southeast region, outside areas where vaccination was recommended for travel [9,10]. Following this outbreak, there is interest in determining how best to manage future YF burden in Brazil by identifying most vulnerable locations to high disease burden.

Vulnerability to infectious diseases has been previously defined in multiple ways, including susceptibility to outbreaks, likelihood of seeing high case burden, or inability to adequately care for observed cases [11]. Definitions of vulnerability to infectious diseases have commonly relied on socioeconomic factors [11–14]. While socioeconomic factors are relevant to the risk of YF and other mosquito-borne diseases, there exist additional relevant factors, notably including various factors of the natural environment [11,15–19]. Mosquito populations are sensitive to climatic factors such as temperature and rainfall [20], and certain ecological habitat types are more hospitable for mosquitoes [15]. Climatic and environmental factors can also impact activities of humans [21] and non-human primates [22], furthering their effects on YF dynamics. Considering ecohydrological conditions such as rainfall or water drainage to predict regions with high vulnerability to YF burden is important when establishing priorities for surveillance and preventative measures. In general, ecohydrological conditions such as altitude and drainage are not practical targets for intervention themselves, but they are beneficial for identifying vulnerable locations to YF burden by representing underlying conditions affecting disease burden. Locations vulnerable to YF burden based on these conditions benefit from improving surveillance or undertaking preventative measures.

Ecohydrological conditions may affect vulnerability to YF by facilitating either sylvatic or urban disease transmission. In Brazil, sylvatic transmission commonly occurs in areas within or adjacent to the Amazon rainforest, near non-human primate reservoir habitats, and it is primarily spread by the *Sabethes* or *Haemagogus* genera of mosquito, which dwell in tree holes. Urban transmission, by contrast, occurs in areas with greater human development, including the coast of Brazil and is primarily spread through the *Aedes* genus of mosquito, which live in areas with standing water. In recent decades, the majority of YF transmission in Brazil occurred through the sylvatic cycle.

A concern in assessing vulnerability is the spatial scale considered. Many studies measuring vulnerability emphasize the vulnerability of a nation or other large spatial region [12,13]. Though it is less commonly done, assessing vulnerability for more localized areas is beneficial for more efficiently targeting preventative measures within a country. In public health practice, undertaking preventative measures, which can include vaccination campaigns or mosquito control, are less practical when applied over entire nations. Targeting smaller, specific locations that are more vulnerable is more resource efficient.

When assessing disease vulnerability among small areal units, it is likely that many locations will not have observed any cases over the time period of observation. This could be because of epidemiologic silence, in which infections may have occurred without being detected by health or surveillance systems [23–25], or because infections did not occur during the time period. A true lack of infections may be due to conditions that decrease disease risk. Without knowing whether cases are not seen due to epidemiologic silence or lack of risk, it is a more conservative approach to assume that locations that did not observe cases are still susceptible. Following this assumption, vulnerability would be estimated for all locations within the region of interest.

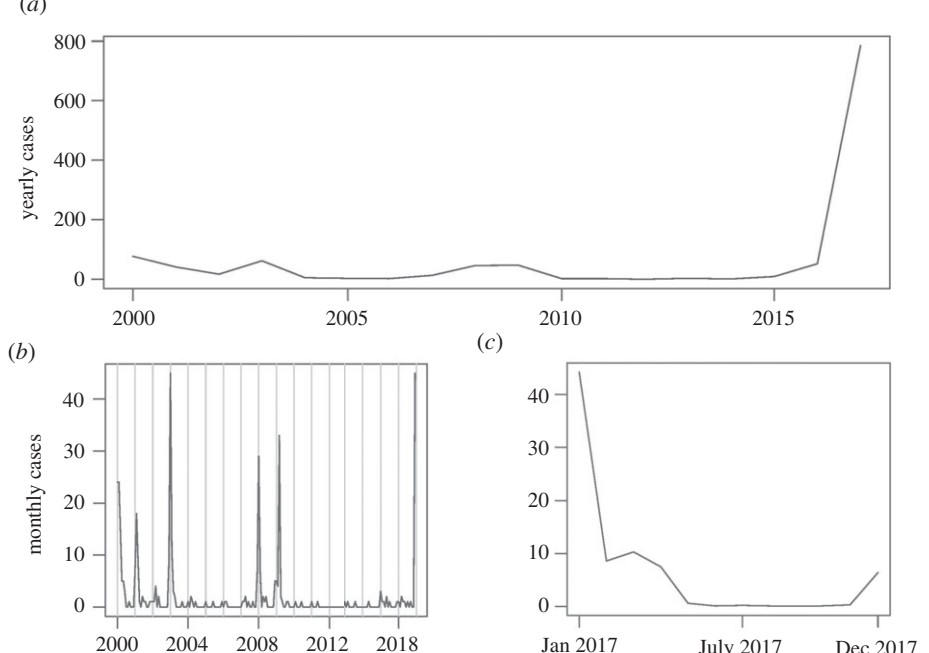

**Figure 1.** Time series of total YF cases in Brazil: (*a*) yearly cases, 2000–2017; (*b*) monthly cases, 2000–2016; (*c*) monthly cases, 2017.

This study aimed to estimate vulnerability to YF burden throughout Brazil using environmental features, in particular assessing the disease niche driven by ecohydrological features sensitive to climate. Vulnerability was defined categorically based on annual incidence and then estimated based on environmental characteristics through a statistical model. Predictors were fit to the vulnerability categories among municipalities where YF was observed, and model results were then extended to include municipalities where YF had not been observed to produce vulnerability estimates by municipality nationwide. The results of this study provide estimates of vulnerability to high YF burden based on environmental characteristics, which allow locations within Brazil that most closely resemble locations with the greatest disease burden to be identified and targeted for public health interventions.

# 2. Method and materials

## 2.1. Study setting

The study region is Brazil, the largest nation in South America both by land area and by population. The nation is subdivided into 5570 municipalities [26], providing the spatial units of analysis. Data were divided into two time periods due to an outbreak that began in December 2016 [8]; this accounts for potential changes in disease dynamics during an outbreak. The first time period, representing vulnerability to YF outside of an outbreak, spanned between 2000 and 2016. The second time period, representing vulnerability to YF during and immediately following an outbreak, spanned 2017. Figure 1 shows the time series of yearly YF cases during these years.

Vulnerability to YF was defined using yearly incidence (cases per 100 000 people annually) for each municipality. Data were analysed by municipality-year, representing each municipality in each year of observation. Analysing yearly burden encompasses seasonal trends throughout the year by including the season of increased YF transmission along with the remainder of the year with less transmission. Vulnerability was defined in three categories: minimal vulnerability (denoted category 0), defined by observing zero cases per year; low vulnerability (denoted category 1), defined by observing up to 10 cases per 100 000 people annually; and high vulnerability (denoted category 2), defined by observing more than 10 cases per 100 000 people annually (table 1). Using 10 cases per 100 000 people provided an intuitive, interpretable cut-off as well as similar sample sizes for low and high vulnerability.

**Table 1.** Vulnerability categories for yearly YF incidence among municipalities where YF or dengue cases were observed in Brazil, 2000–2017.

| category number | category name | incidence definition | number of municipality-years, 2000–2016 | number of municipalities, 2017 |
|---|---|---|---|---|
| 0 | minimal vulnerability | 0 cases per 100 000 people | 15 648 | 770 |
| 1 | low vulnerability | 0–10 cases per 100 000 people | 109 | 75 |
| 2 | high vulnerability | >10 cases per 100 000 people | 86 | 87 |

## 2.2. Data sources

### 2.2.1. Yellow fever incidence

YF case data were provided by the Pan American Health Organization and the Brazilian Ministry of Health. Case data were collected through active surveillance activities, with healthcare providers reporting confirmed cases to the Ministry of Health. Case reporting of YF is mandatory, and both human and non-human primate cases are investigated following case reports. The date of report and the municipality associated with each case are included. Case data were available between January 2000 and March 2018, though only data to 2017 were used in analysis to include only complete years.

Population data for Brazilian municipalities are publicly available through the Brazilian Institute for Geography and Statistics [26]. Annual population estimates are available for all years between 2000 and 2017, with the exceptions of 2007 and 2010, where the arithmetic mean of the two adjacent years was used to singly impute missing values. With population data and case data, annual incidence values per 100 000 people were calculated for each municipality-year and then converted into vulnerability categories defined in table 1.

### 2.2.2. Environmental predictors

Environmental characteristics were included to estimate vulnerability to YF burden. Candidate predictors were selected based on connections with YF risk and then selected for inclusion in the predictive model based on predictive ability. The candidate predictors chosen represent environmental characteristics that have previously been associated with YF or other mosquito-borne diseases. Vulnerability based on such conditions can identify regions within Brazil in greatest need of YF surveillance and prevention based on the natural environment. Furthermore, the candidate predictors in this study include various ecohydrological factors such as altitude and water drainage, which are relevant to YF burden, but less extensively studied.

Geospatial data (in the form of shapefiles) for Brazilian municipalities were available from the Database of Global Administrative Areas v. 3.6 [27]. The shapefiles, which are current as of 2019, contain land area in square kilometres for each municipality. Using these and the population data described previously, population density per square kilometre was calculated, in thousands, for each municipality to represent urbanization.

Precipitation and temperature have been previously linked to risk of YF [28–31]. Precipitation and temperature data are publicly available through the Modern-Era Retrospective Analysis for Research and Applications, v. 2, through the United States National Aeronautics and Space Administration [32]. Both are available in 0.5-degree by 0.625-degree grids, represented as millimetres of precipitation per hour and ambient temperature in Celsius [32], and are available for the entire time period of this study. To be compared with annual YF incidence, precipitation data were aggregated to represent the percentage of days for each municipality each year from 2000 to 2017 that experienced any rainfall. Because many urban mosquito species can lay eggs in small amounts of water [33], this variable represents the frequency of opportunities for mosquito breeding grounds to be created throughout the year. Similarly, sylvatic mosquito species such as *Haemagogus* rely on rainfall, with oviposition and larval development relating to rainfall [34]. Temperature data were aggregated to represent annual

mean temperature. Temperature has been shown previously to relate to mosquito life cycles as well as ability to incubate viruses [20]. Grids were matched to municipalities using a spatial join in ArcGIS Pro v. 2.2.0 [35]. Precipitation and temperature were considered as linear and quadratic predictors in model fitting.

Standing water provides opportunities for mosquito breeding [33] and has been shown to relate to YF or other mosquito-borne viruses through river drainage [36], storm drains [37] and irrigation systems [38], primarily in urban settings. Drainage density, defined as the ratio of the length of rivers and streams to land area, was modelled for Brazil accounting for elevation and known hydrological features in the area. This represents relative amounts of water movement through municipalities, which may create areas of standing water to serve as mosquito breeding grounds. Drainage density was estimated for water basins as of 2017, where basins are defined to have a minimum area of 1000 km². Average drainage density, weighted by proportion of land overlap between basins and municipalities, was computed for municipalities using a spatial join. Drainage density is assumed to remain constant over the entire study period.

Average elevation in metres, which has also been shown to relate to YF risk, in that lower altitude regions were found to be more likely to observe YF cases [15], is publicly available through the Advanced Spaceborne Thermal Emission and Reflection Radiometer Digital Elevation Model, v. 2, from the United States National Aeronautics and Space Administration and Japan's Ministry of Economy, Trade, and Industry [39]. These data are available in 30 by 30 m grids representing 2011, which were matched to municipalities using a spatial join to represent average elevation in each municipality. Elevation is assumed not to change throughout the study time period.

Vegetation has been previously related to YF risk [40]; low vegetation is indicative of human development and defines areas of urban transmission, while vegetation in sylvatic areas represents habitat opportunities for tree-dwelling mosquitoes. Vegetation data were available from MODIS in the form of the normalized difference vegetation index (NDVI). Monthly data were collected for each municipality centroid through the R package 'MODISTools' [41] and then averaged to represent yearly vegetation density.

Terrestrial biomes, which have been shown to relate to YF risk [15], are publicly available from the World Wildlife Foundation [42]. These biomes can represent access to forested areas. A total of six biomes, reflecting ecohydrological and geomorphological factors, are present in Brazil: deserts and xeric shrublands; flooded grasslands and savannahs; mangroves; tropical and subtropical dry broadleaf forests; tropical and subtropical grasslands, savannahs and shrublands; and tropical and subtropical moist broadleaf forests [42] (electronic supplementary material, figure S1). Biomes, which were reported for 2012, are assumed to remain constant throughout the entire study period and were matched to municipalities to assign to each municipality the biome with the highest percentage of land cover. Three biomes (flooded grasslands and savannahs, mangroves, tropical and subtropical dry broadleaf forests) were present in very few municipalities and therefore collapsed into a single category with the least frequent of the remaining three (deserts and xeric shrublands). The four collapsed biomes as well as tropical and subtropical grasslands, savannahs and shrublands were considered as two indicator predictors, leaving tropical and subtropical moist broadleaf forests, the most common biome, as the reference for analysis.

Another factor considered in assessing vulnerability is occurrence of the disease in nearby locations, which can account for movements and disease transmission among humans, non-human primates and mosquitoes. Occurrence of at least one case of YF in any neighbouring municipality was included as a binary predictor. Two definitions of neighbour were considered: (i) sharing a land border and (ii) having a centroid within a 50 km radius from a given municipality's centroid.

## 2.3. Data analysis

In order to reduce uncertainty of whether municipalities that did not observe YF cases were epidemiologically silent or truly observed no cases, only the 466 municipalities where at least one case was seen between January 2000 and March 2018 as well as 466 randomly selected municipalities where YF was not seen, but dengue was seen in 2006, were included in analysis (figure 2). The municipalities where dengue was seen, serving as pseudo-absence municipalities, were publicly available from Brazil's Ministry of Health, through its Unified Health System [43]. These 932 municipalities (466 where YF was observed + 466 pseudo-absence municipalities) were used in data analysis for both the 2000–2016 time period and for 2017; this allowed all three vulnerability categories to be present in both datasets during analysis (table 1).

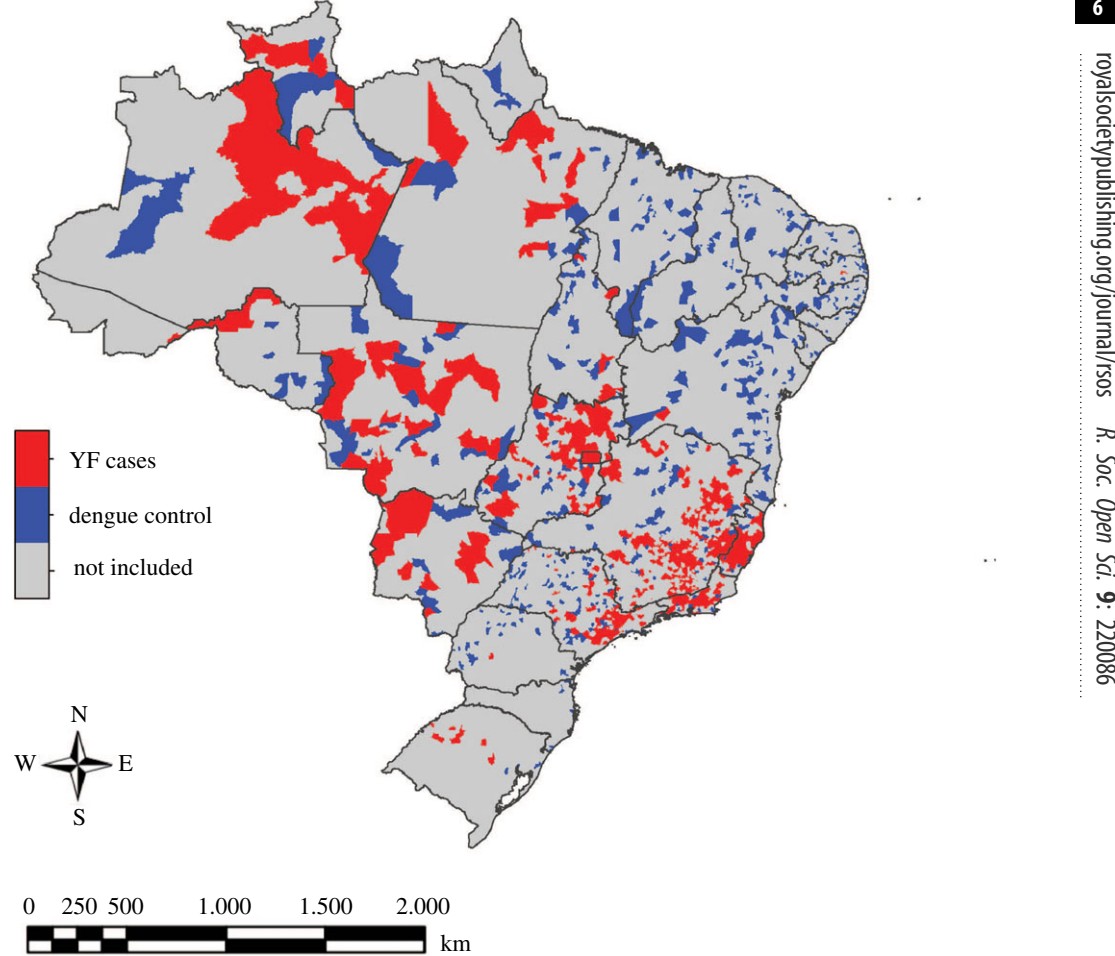

**Figure 2.** Municipalities of Brazil where any YF cases or dengue cases were seen between January 2000 and March 2018. The highlighted municipalities' vulnerability and environmental data were used in model fitting for both time periods.

Inclusion of only municipalities where at least one YF or dengue case was observed between January 2000 and March 2018 when analysing data for both time periods assures that all three vulnerability categories are present in the data. For both time periods, excluding municipalities where no YF or dengue cases were ever seen increases confidence that the absence of cases in the data represents a true absence of cases rather than an artefact of epidemiologic silence because observing at least one YF or dengue case increases confidence that other cases were able to be detected in those municipalities. This does, however, assume constant ability to detect cases in these municipalities during the time period of observation.

The outcome representing vulnerability to YF (minimal, low and high vulnerability categories) by municipality-year was fit to a cumulative logit regression model to produce estimated probabilities associated with the three ordinal categories. Cumulative logit models carry the advantage of producing estimated probabilities for ordered categories, such as the vulnerability categories defined in this study. Two separate models were developed for the two time periods described previously. Both analyses were conducted separately over the 932 municipalities described previously.

The aim of model fitting was to find the set of predictors that produces the most accurate fits of the vulnerability categories. The previously described predictors were candidate predictors for the model, with inclusion reliant on improvement of predictive ability. A total of 863 parameter combinations were considered, which included all combinations of inclusion and exclusion, omitting combinations that included the quadratic term for percentage of rainy days or temperature without including the linear term, combinations that included only one biome indicator, and combinations that include both indicator variables for YF presence in neighbouring municipalities.

Closest fit was determined based on receiver operator characteristic (ROC) curves and associated areas under the curve (AUC). ROC curves for binary analyses can be extended to create two curves:

one for discriminating between minimal vulnerability versus low or high vulnerability and one for discriminating between minimal or low vulnerability versus high vulnerability [44]. The parametrization providing the highest total of the two AUCs was selected as the best-fitting model for each time period. Models were run using the 'MASS' package [45] in R v. 3.6.0 [46].

# 3. Results

All candidate variables described previously, with summary statistics shown in table 2, were eligible for inclusion into the model for either time period, with the caveat that only one of the two definitions for neighbouring municipality, if any, can be included. Visual representations of environmental predictors, showing spatial heterogeneity in their values, can be found in the electronic supplementary material, figure S2.

## 3.1. Model fit and validation

The model using data from 2000 to 2016 found that the best-fitting predictor set consisted of percentage of days with rain as a linear term, population density, vegetation and case presence in municipalities within 50 km (table 3). The total AUC was 1.97.

The model using data from 2017 found that that best-fitting predictor set consisted of percentage of days with rain as a linear term, mean temperature as a linear and quadratic term, population density, elevation, biome and case presence in bordering municipalities (table 3). The total AUC for this model was 1.80.

The log odds ratios in table 3 represent the association between each predictor and observing a higher vulnerability category (comparing minimal with low vulnerability and low with high vulnerability using the same value). The fitted model for 2000–2016 shows that a lower percentage of days with precipitation associated with higher log odds associated with observing a higher vulnerability category. Lower population density was also associated with higher log odds of observing higher vulnerability. Higher NDVI values, indicative of greater vegetation density, were associated with higher vulnerability. These associations were low in magnitude and non-significant. Observing YF in a municipality within 50 km was associated with higher log odds of observing higher vulnerability categories.

The model using 2017 data shows a significant association between frequency of rain and vulnerability, with a lower percentage of days with precipitation associated with a greater risk of observing a higher vulnerability category. A parabolic trend was seen between mean temperature and vulnerability, with the highest log odds observed near 22°C. Municipalities with lower population density saw higher log odds of observing higher vulnerability, and municipalities with higher elevation saw higher log odds, though with low magnitude. Being located in any other biomes was associated with lower log odds of high vulnerability compared with tropical and subtropical moist broadleaf forests, and observing cases in bordering municipalities was associated with higher vulnerability (table 3).

## 3.2. Estimated vulnerability

The log odds ratios presented in table 3 were used to generate fitted probabilities associated with each of the three vulnerability categories, which sum to one by design. During 2000–2016, all municipalities showed low probability (0.00–0.06) of observing both moderate and high vulnerability (figure 3b,c). It follows that all municipalities showed high probability (0.89–1.00) of observing minimal probability (figure 3a). The 25 municipalities with the highest probabilities (0.03–0.06) of observing high vulnerability were seen in the North and Central-West regions.

In 2017, approximately 83% of municipalities were estimated to have a probability of observing minimal vulnerability to YF greater than 0.9 (figure 3d). The 256 municipalities with the highest fitted probabilities (0.24–0.25) for observing low vulnerability and 48 municipalities with the highest fitted probabilities (0.45–0.58) for observing high vulnerability were primarily seen in the Southeast region as well as three municipalities in Bahia state (figure 3e,f).

Priorities for preventative measures based on these fitted probabilities can be made by defining cut-offs in the fitted probabilities or identifying a specific number of municipalities to target. For example, prioritizing only municipalities with fitted probabilities associated with high vulnerability above 0.5 would lead to no municipalities being prioritized based on 2000–2016 data and 16 municipalities

**Table 2.** Descriptive statistics of predictor variables for YF vulnerability models, 2000–2017. Mean and standard deviation values are shown for continuous predictors, and counts of positive values with percentages are shown for binary variables.

| predictor | mean (N) | s.d. (%) | min | 25% | 50% | 75% | max |
|---|---|---|---|---|---|---|---|
| **2000–2016** | | | | | | | |
| % rain days | 47.1 | 12.9 | 10.4 | 39.7 | 45.8 | 51.8 | 100 |
| temperature | 23.1 | 2.5 | 16.8 | 21.1 | 22.9 | 25.1 | 29.5 |
| population density (1000) | 0.195 | 0.826 | 0.000 | 0.012 | 0.029 | 0.076 | 13.267 |
| elevation | 508.289 | 285.961 | 6.438 | 289.931 | 508.013 | 734.937 | 1414.086 |
| vegetation | 0.5925 | 0.128 | −0.300 | 0.524 | 0.602 | 0.675 | 0.902 |
| drainage | 0.091 | 0.016 | 0.055 | 0.082 | 0.089 | 0.095 | 0.343 |
| biome—tropical and subtropical grasslands, savannahs and shrublands | 3910 | 24.7 | 0 | 0 | 0 | 0 | 1 |
| biome—other | 2550 | 16.1 | 0 | 0 | 0 | 0 | 1 |
| cases within 50 km | 835 | 5.3 | 0 | 0 | 0 | 0 | 1 |
| cases bordering | 364 | 2.3 | 0 | 0 | 0 | 0 | 1 |
| **2017** | | | | | | | |
| % rain days | 44.0 | 12.6 | 17.8 | 36.3 | 43.3 | 47.9 | 98.6 |
| temperature | 23.3 | 2.6 | 18.4 | 21.1 | 22.9 | 25.3 | 28.7 |
| population density (1000) | 0.217 | 0.898 | 0.000 | 0.013 | 0.031 | 0.083 | 13.353 |
| elevation | 508.271 | 286.106 | 6.438 | 289.761 | 506.306 | 733.973 | 1414.086 |
| vegetation | 0.592 | 0.138 | −0.300 | 0.523 | 0.604 | 0.684 | 0.890 |
| drainage | 0.091 | 0.016 | 0.055 | 0.082 | 0.089 | 0.095 | 0.343 |
| biome—tropical and subtropical grasslands, savannahs and shrublands | 230 | 24.7 | 0 | 0 | 0 | 0 | 1 |
| biome—other | 150 | 16.1 | 0 | 0 | 0 | 0 | 1 |
| cases within 50 km | 449 | 48.2 | 0 | 0 | 0 | 1 | 1 |
| cases bordering | 261 | 28.0 | 0 | 0 | 0 | 1 | 1 |

**Table 3.** Parameter estimates, representing log odds ratios associated with observing a higher category of vulnerability to YF incidence in Brazil, 2000–2016 and 2017. Empty cells indicate that a predictor was not included in the best-fitting model.

| predictor | estimate (95% CI), 2000–2016 | estimate (95% CI), 2017 |
|---|---|---|
| per cent days with rain | −0.585 (−1.459, 0.289) | −6.034 (−6.965, −5.687) |
| per cent days with rain$^2$ | | |
| mean temperature | | 1.298 (1.102, 1.494) |
| mean temperature$^2$ | | −0.029 (−0.037, −0.021) |
| population density | −0.331 (−1.090, 0.428) | −0.234 (−0.495, 0.027) |
| elevation | | 0.000 (−0.001, 0.001) |
| vegetation | 0.037 (−1.088, 1.162) | |
| drainage density | | |
| biome—tropical and subtropical grasslands, savannahs and shrublands | | −0.682 (−1.313, −0.051) |
| biome—other | | −3.279 (−5.167, −1.392) |
| YF in neighbouring municipality (border[a]) | | 2.359 (1.844, 2.874) |
| YF in neighbouring municipality (50 km[b]) | 21.462 (21.084, 21.840) | |
| Intercept 0\|1 | 22.337 (21.959, 22.715) | 14.180 (14.156, 14.204) |
| Intercept 1\|2 | 23.315 (22.903, 23.727) | 15.197 (14.979, 15.415) |

[a]Refers to observing any YF cases in a municipality sharing a land border.
[b]Refers to observing any YF cases in a municipality with centroids within 50 km.

being prioritized based on 2017 data. Alternatively, identifying, for example, the 100 municipalities with highest probabilities associated with high vulnerability would lead to cut-offs of 0.02 using the 2000–2016 data and 0.41 using the 2017 data. Priorities can also be set including the probabilities for the other two categories, such as prioritizing municipalities with fitted probabilities associated with low vulnerability below a threshold or applying weights to the probabilities for low and high vulnerability to develop a priority score.

## 3.3. Sensitivity analyses for disease pattern critical threshold and predictor influence

Full methods and results of sensitivity analyses can be found in the electronic supplementary material. Briefly, a sensitivity analysis was conducted to determine if model results are sensitive to the value of incidence used as a cut point between the low and high vulnerability categories. Model fit was compared for different cut points. Using the 2000–2016 model, cut points between 9 and 18 as well as between 25 and 33 yielded the highest AUC values. Using the 2017 model, cut-offs between 9 and 12 as well as 30 yielded the highest AUC values. The chosen cut-off of 10 was shown to be both intuitive for interpretation and suitable for model fit.

A global sensitivity analysis [47,48] was conducted to determine if the parameter set was primarily driven by any single input variables and for assessing variable interdependence. Sobol indices [49,50] were adapted to the cumulative logit model to determine the most influential variables contributing to the probabilities associated with being in each of the three vulnerability categories. The results of these analyses indicate if any predictors have greatest relative influence on vulnerability to YF compared with the others. The indicator variable for cases within 50 km was the most influential predictor for the 2000–2016 model (electronic supplementary material, table S1; figure 4). The mean temperature was the most influential predictor for the 2017 model, through both its linear and quadratic terms (electronic supplementary material, table S2; figure 4). Since these were considered the most influential predictors of vulnerability categories, in a situation where only some, but not all, of the predictors used in this study are able to be ascertained, it is beneficial to prioritize ascertaining information pertaining to these predictors.

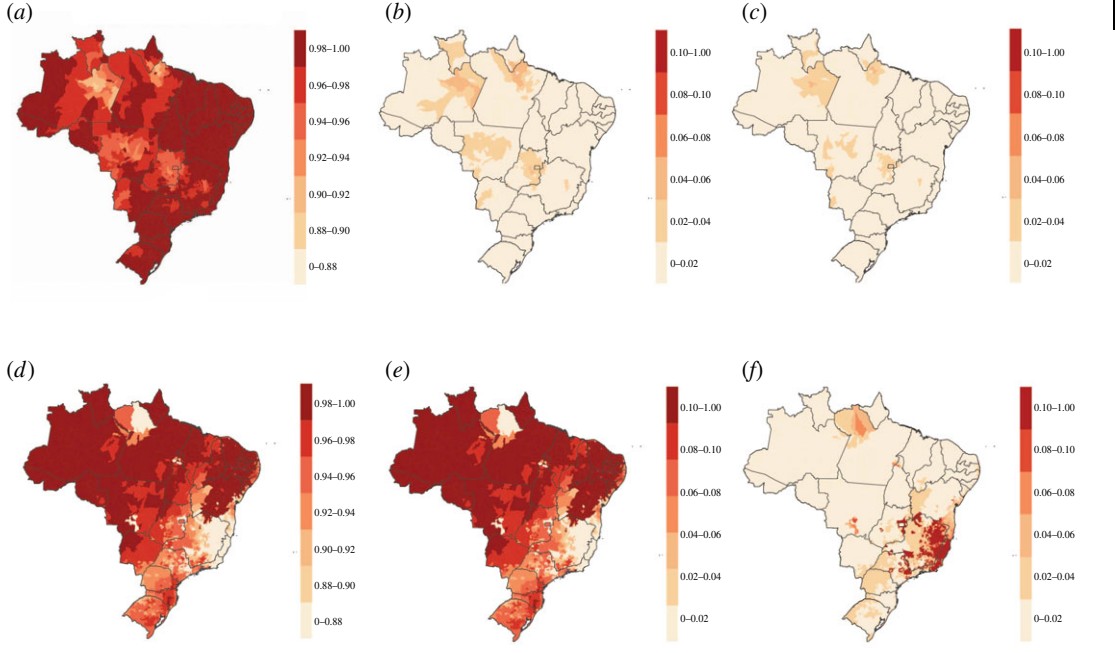

**Figure 3.** Estimated probabilities associated with vulnerability categories for YF burden in Brazil. Estimates for 2000–2016: (*a*) minimal, (*b*) low and (*c*) high vulnerability as well as 2017: (*d*) minimal, (*e*) low and (*f*) high vulnerability categories were predicted with a cumulative logit model using environmental predictors. Scales across graphs are inconsistent to show spatial heterogeneity within predictions for each category; maps using the same colour scale are found in the electronic supplementary material, figure S3.

## 4. Discussion

This study aimed to identify municipalities of Brazil with the highest vulnerability to YF burden based on environmental and hydrological conditions, defined through categories of yearly incidence. Best-fitting statistical models were estimated using environmental predictors, highlighting ecohydrological characteristics in particular. The model fitting process identified environmental conditions that best predict YF vulnerability, through both inclusion in the model as well as magnitude and significance of association. Model results were used to estimate vulnerability nationwide. This emphasized the locations with the greatest need for preventative measures due to innate characteristics. Targeting initiation or continuation of interventions to the most vulnerable locations and identifying those with minimal YF risk can serve to allocate limited resources. Using different criteria to prioritize municipalities, such as different cut-offs of probabilities for certain categories, may lead to different prioritizations and therefore resource allocations. The model from 2000 to 2016 shows all municipalities to have a low probability of observing high vulnerability. The model using 2017 data shows that municipalities with higher fitted probabilities associated with high vulnerability were primarily located in southeastern Brazil.

The predictor sets between the two time periods differed, with the model for the 2000–2016 time period having fewer included predictors, most of which showed associations that were non-significant and of small magnitude. Three of the four parameters, rainfall, population density and vegetation, are indicative of sylvatic transmission, as areas of marginally higher vegetation and lower population density being more vulnerable to observing high YF burden. The model for 2017 implies some sylvatic transmission through a negative association with population density. The other predictors for this model could be indicative of either sylvatic or urban transmission.

Occurrence of YF in nearby locations was an influential parameter in the predictor set for vulnerability during the 2000–2016 time period. It was the only parameter in the 2000–2016 model that was statistically significant and of higher magnitude. This finding suggests that there is a spatial relationship in YF transmission. This is supported in previous work [51] and by the data; visual examination of the locations where YF was seen (figure 2) shows that municipalities where cases were seen tend to be near each other. This may emphasize the importance of local human activities on YF dynamics in Brazil or result from nearby locations having similar climate conditions (electronic supplementary material, figure S2).

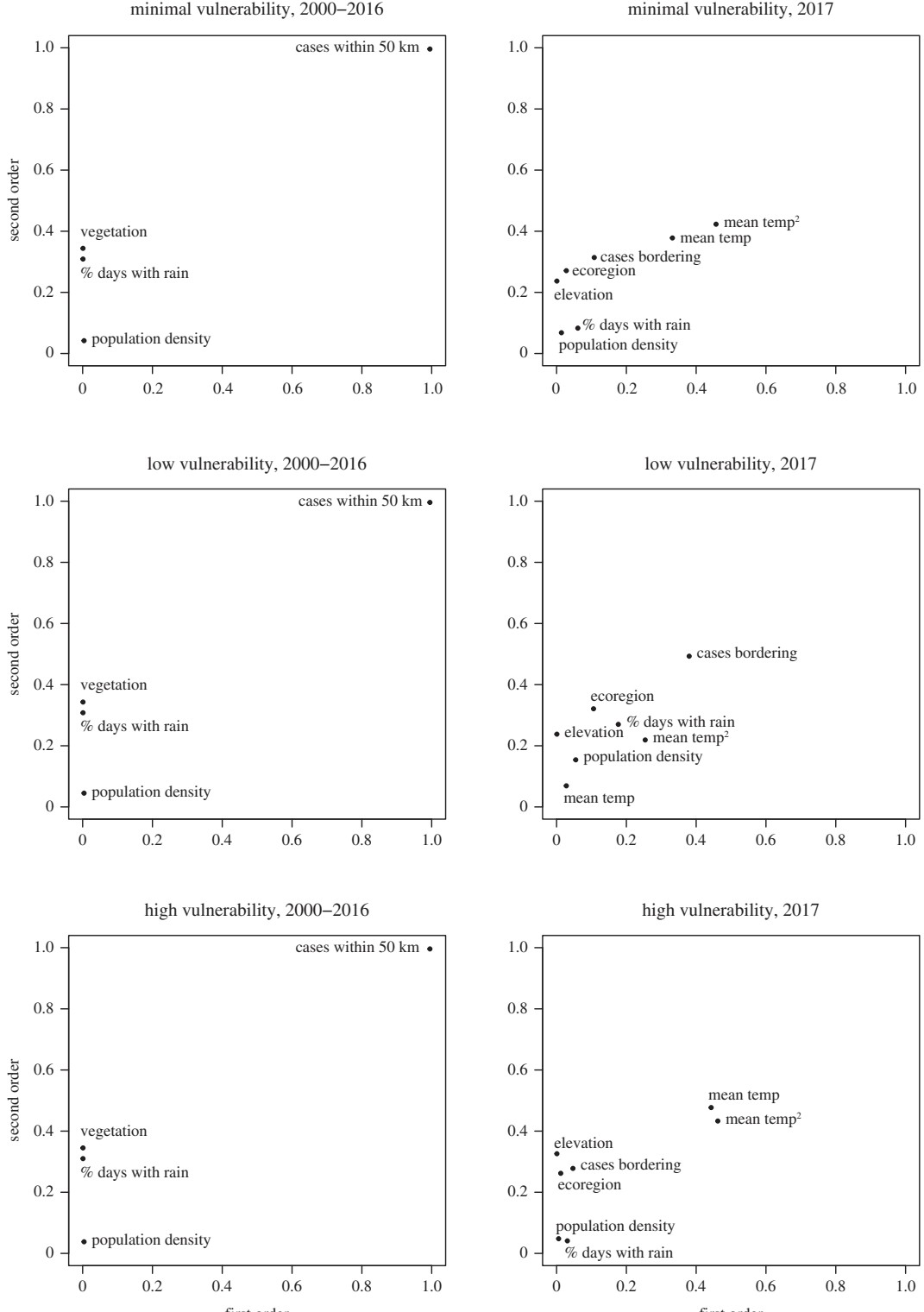

**Figure 4.** First- and second-order Sobol indices for model predictors for the models fit using 2000–2016 (left column) and 2017 (right column) data. Higher values of Sobol indices represent greater relative influence in model prediction as a function of their independent variability and joint variability for first- and second-order index, respectively.

The resulting models in this study were applied to all municipalities in order to assess nationwide vulnerability, allowing most vulnerable municipalities to be identified. Similar extensions of results have been done in other studies by applying model results to locations outside of the dataset used [52], while others avoided extrapolating and applied results only to locations providing data [53]. Within this study, the extrapolation used to produce national estimates relies on the assumptions of

comparable mechanisms existing between environmental predictors and disease risk nationwide. Doing so also assumes that all municipalities of Brazil are capable of observing YF cases, assuming epidemiologic silence rather than a true inability to observe cases when cases are not seen. This is important, however, because it considers vulnerability in all locations, including those that may not have previously been prioritized for surveillance and prevention.

This study offers insight into YF vulnerability at a high spatial resolution across the entire nation of Brazil. While other studies have compared nations for their disease risks [13,54], this study identifies the municipalities within one nation with characteristics that indicate having greater vulnerability to YF burden. Other studies using small areal units for similar purposes commonly focus on a more localized region [55–57] rather than a large nation.

Previous works addressing vulnerability to infectious diseases have used socioeconomic factors such as the history of infection [11] or through the connection between underserved populations and disease risk [58,59]. This study instead focuses on environmental factors, representing a different set of disease determinants. For YF and other mosquito-borne diseases, previous studies have considered various environmental determinants and predictors of incidence, including temperature [60,61], precipitation [62,63], elevation [64], biomes [15] and vegetation [65]. Many of these or similar predictors were included in this study in order to estimate vulnerability to YF in Brazil.

The use of a global sensitivity analysis for identifying influential model predictors is beneficial for contextualizing results and determining if any particular predictors among the predictor set are of particular importance in estimating YF vulnerability. Showing the strongest predictors can offer insight into most effective implementation of preventative measures or identify predictors in which results are sensitive to measurement errors. For example, observing cases in nearby municipalities was the most influential predictor for 2000–2016, and frequency of rainfall was the most influential for 2017. The results of this study therefore may be most sensitive to either of these predictors.

## 4.1. Implications for preparedness

The results of this study identified municipalities that are expected to be more likely to experience high yearly burden of YF based on categories of vulnerability, making them the most beneficial targets for prevention strategies for YF. Municipalities with the highest predicted probabilities for the highest vulnerability category are those that are most likely to experience high burden, whether by high case counts at once or by frequent occurrences of cases. In either situation, preventative measures would be most important in these municipalities to prevent YF infection. Such potential strategies for municipalities classified as high vulnerable for YF can include vaccination campaigns [66], establishment of early warning systems [67,68], or vector control strategies [69,70], all of which have been employed in the past.

Applying the model from each time period to inform preparedness currently would yield different priorities for prevention measures. The 2018–2019, 2019–2020 and 2020–2021 seasons saw much lower YF case counts in Brazil, which were found primarily in the South region of the country [71], which is distinct from where most cases were seen in the YF data used in this study. Considering estimates from both models in tandem or obtaining current data to update the model would be recommended for applying the results of this study to practice.

Estimating vulnerability in the municipalities outside the 932 used for model fitting shows municipalities where high categorical vulnerability to YF is more probable compared with others, even if it has not been seen previously and if predicted probability is still low. While the model using 2000–2016 data showed low probability of observing the highest vulnerability category nationwide (electronic supplementary material, figure S3), variation in fitted probabilities within this low range of probabilities was observed (figure 3).

## 4.2. Limitations and future directions

Focusing on environmental predictors of vulnerability neglects other relevant factors that may contribute to YF burden. These can include many important human-related characteristics such as vaccination [52] or socioeconomic status [19,72]. Based on estimated vaccination in Brazil as of 2016 from Shearer *et al.* [66], locations that were considered most vulnerable from the 2000–2016 model were typically those with high vaccination, and those considered most vulnerable from the 2017 model had lower vaccination (electronic supplementary material, figure S5). Instead, this study focused on predictors of vulnerability primarily based on natural environmental features (except for population density), most of which cannot easily be targeted for direct intervention. Doing so highlights location-specific

environmental determinants of YF burden. Focusing on climatological and ecohydrological features underpinning climate sensitivity of YF emphasizes locations where human intervention is most necessary if the importance of these features is high.

In estimating vulnerability among the municipalities where YF cases were not observed, it is assumed that these municipalities are susceptible to YF cases. Other possibilities, however, exist. These municipalities may not have observed YF cases due to undetected cases in relation to poor surveillance [24,73], a true absence of cases by chance or a true absence of cases due to notable differences in characteristics not accounted for in this study, such as human intervention [66,72]. This cannot be determined within the scope of this study; assuming that all municipalities are susceptible to YF infection is the most cautious of these three possibilities.

The distinct differences in predictor sets and in fitted vulnerability between the two time periods, before and after the start of a major outbreak, show a limitation of the models' ability to predict major changes to YF dynamics. Each model had high ability to discriminate across the categories of vulnerability, but did not predict a geographic shift in burden. This limits the applicability of the models, where they will apply to data within their own time period and would require updates following major changes in geography. The 2016–2018 YF outbreak in Brazil largely occurred in areas that were not considered vulnerable by the previous years' model, which similarly had low vaccination coverage [66]. To maximize usefulness of the analyses presented here, updating the models to reflect current trends would be needed to identify the currently most vulnerable locations.

Results from the model using 2000–2016 data show that the occurrence of YF cases within 50 km was a strong predictor of YF vulnerability, with other predictors showing small magnitude and statistical non-significance. This limits the model's ability to show that sharing similar environmental characteristics of high-burden locations places another location as vulnerable to high YF burden. Instead, this model may better show the spatial dependence of YF, where municipalities with high burden tend to be near each other. Nevertheless, the 2017 model shows stronger connections between environmental conditions and YF vulnerability, and the 2000–2016 model is beneficial for identifying the strength of the spatial dependence for preparedness purposes.

In addition to the preventative measures that can be informed by the results of this study, future research topics can benefit from these results. This study incorporates factors of the natural environment to predict vulnerability to YF burden. These predicted vulnerability categories could prove useful when predicting immediate risk of YF burden but more realistically they are useful for long-term disease risk management. Other studies can investigate other facets of total vulnerability to YF not pursued in this study. Doing so would allow various aspects of vulnerability to be examined and potentially combined, for instance the importance of dynamic social factors such as mobility in comparison with environmental factors.

# 5. Conclusion

Based on environmental predictors including biomes, elevation, temperature and precipitation in addition to spatial proximity, several municipalities within Brazil were identified as those that are most or least likely to observe high categorical vulnerability to YF. These municipalities were identified through having environmental characteristics similar to those where greatest YF burden was seen. Mean temperature and presence of YF cases in nearby municipalities were found to be influential variables in the generation of YF predictions. These results can be used to inform ongoing preparedness strategies in Brazil for future YF burden, including healthcare capacity, mosquito control or risk communication, particularly when climate change scenarios are considered. Knowledge of environmental factors predicting disease burden is crucial for preparedness because human activities cannot immediately control them, but can adapt to them in a proactive ecosystem health perspective.

Ethics. This article does not present research with ethical considerations.

Data accessibility. The R code and data used for this study are stored in GitHub (github.com/jlservadio/YFVuln) and have been archived within the Zenodo repository: zenodo.org/record/5976037.

Authors' contributions. J.L.S.: conceptualization, data curation, formal analysis, funding acquisition, methodology, validation, visualization, writing—original draft and writing—review and editing; C.M.-Z.: conceptualization, data curation, methodology and writing—review and editing; M.C.: conceptualization, data curation, methodology, project administration, supervision and writing—review and editing.

All authors gave final approval for publication and agreed to be held accountable for the work performed therein.

Competing interests. We declare we have no competing interests.

Funding. This funding was funded through a consulting contract through the Pan American Health Organization as well as the Doctoral Dissertation Fellowship from the University of Minnesota. M.C. acknowledges funding from the High-talent Pengcheng Peacock Scheme (B Talents) of the Shenzhen Government, the Talent Start-up Fudnding of Tsinghua SIGS, and funding from the Ministry of Science and Technology of China ("Climate and Anthropogenic Impacts on Coastal Ecosystems: Keystone Species-Habitat and Blue Carbon Feedbacks").

Acknowledgements. The authors thank the Pan American Health Organization and the Brazilian Ministry of Health for providing the YF data used for this study. The authors also thank Fernando Nardi and Antonio Annis (University for Foreigners of Perugia, Italy) for providing the drainage density data.

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
