## [Peer Review File · Royal Society Open Science]

Review History

RSOS-211305.R0 (Original submission)

Review form: Reviewer 1

Is the manuscript scientifically sound in its present form?

Yes

Are the interpretations and conclusions justified by the results?

Yes

Is the language acceptable?

Yes

Do you have any ethical concerns with this paper?

No

Have you any concerns about statistical analyses in this paper?

No

Recommendation?

Accept with minor revision (please list in comments)

Comments to the Author(s)

This paper describes a modelling study with the aims of predicting vulnerability to Yellow Fever (YF) in Brazilian municipalities, based on environmental factors and existing records of the disease. It presents adequate methodology, it is novel and the conclusions are supported by the results. My personal view, however, is that the overall presentation and discussion of the results are somewhat distant from the local dynamics of YF epidemiology in Brazil. Vaccination plays a major role in preventing YF cases, and a low coverage was probably the main determinant of the 2017/2018 epidemic in Minas Gerais. In addition, it would be interesting to improve the discussion on the differences between sylvatic and urban transmission of YF in Brazil, as the paper is centered in environmental determinants. Brazil is endemic for sylvatic YF, and even with the high numbers of the 2017/2018 epidemic, it was not considered as an urban spillover, especially because of the vector species involved in local transmission. Despite the importance of some environmental predictors to vector dynamics, vector's species names are never mentioned in the text. My comments below are attempts to improve the discussion, and do not invalidate the study's main results. Therefore, I believe that a minor revision of the manuscript should be enough for achieving enough quality for publication.

Specific comments and suggestions to the authors:

Abstract:

L6-9: I don't fully agree with the first sentence. I would say that the reasons behind the lack of reported cases in many Brazilian localities are more related with vaccination coverage than with environmental determinants.

L25-26: "...in western and southeastern municipalities..." Brazil is officially divided in five regions: North, Northeast, Southeast, Mid-West and South. Using their recognized names throughout the text would improve interpretation of results and maps.

Introduction:

This section could be improved with a clear justification of why the study was performed in Brazil, and an introduction of the local epidemiology of YF and its dynamics.

Methods:

Do you have any particular reason for not including temperature in the vulnerability models? It is mentioned in the text that it is important for mosquito and YF dynamics, but not included as a variable.

I believe that, because of the way ecoregions were treated in the model, they should be treated as a study limitation and more explored in the discussion. The reasons for making it a binary variable are not clear in the text. I see two potential problems here: 1) from the map in FigS1, I see that you are using what WWF calls biomes and not ecoregions - indeed they do resemble the distribution of the six Brazilian biomes: Amazon, Cerrado, Atlantic Forest, Caatinga, Pantanal, Pampas (maps can be found at IBGE and MMA - Brazilian Ministry of Environment). 2) merging different biomes into two categories (FigS2) might hinder your ability to distinguish their specific effects in the disease outcome.

If you are willing to re-run the analysis, I would suggest treating the six biomes separately. If a 6-level categorical variable is not adequate for your method, I would suggest breaking them down into 6 separate binary variables, to assess the effect of each biome in YF. That can be achieved by reclassifying the values of one biome to 1 and the remaining to 0, for each of the biome classes.

I do not see that as a mandatory correction - you could otherwise keep your current results and be clearer of their limitations - but it would certainly improve your findings and conclusions.

Nevertheless, a great and relevant work.

Review form: Reviewer 2

Is the manuscript scientifically sound in its present form?

Yes

Are the interpretations and conclusions justified by the results?

No

Is the language acceptable?

Yes

Do you have any ethical concerns with this paper?

No

Have you any concerns about statistical analyses in this paper?

No

Recommendation?

Major revision is needed (please make suggestions in comments)

Comments to the Author(s)

The authors here address an important question, what environmental determinates predict population vulnerability to high YF incidence. While the paper is very well written, and commendably the code is located on github, there are several methodological issues with the analysis and interpretation.

There is an omission of several covariates (EVI, temperature, seasonality, landcover, etc) that have already shown to be important correlates with YF activity, and inclusion of ones not relevant for sylvatic YF (drainage density). These should be expanded considerably. Additionally, while the paper is on the "environmental determinants" there is no reasoning behind omitting additional covariates which may determine the reporting of YF (socio-economics, access to sylvatic habitats, non-human primate presence). The restriction of the dataset to 447 locations, rather than employing pseudoabsence points or another method has led to what appears to be a substantial degree of overfitting, though as they do not provide an interpretable metric of model performance it is hard to say exactly how much. While the authors accurately note that the absence of evidence is not evidence of absence, restricting the dataset without compensating appears to have led to an uninformative model. This is highlighted in the results, as it appears that most of the covariates used in the models are insignificant, and so large portions of the results and discussion are not appropriate.

In a research area that already has numerous papers published on environmental/socio-economic suitability of South America to YF, these omissions are inappropriate. I would strongly recommend that the authors revisit their covariates used, expand them, and employ pseudo-absences in their modelling process, as well as report an interpretable metric on the model fit, only then will we know if the work undertaken here has expanded the field of knowledge on the subject.

Specific comments

Methods

Page 8 Line 20-25: “precipitation data were aggregated to represent the percentage of days for each municipality each year from 2000 through 2017 that experienced any rainfall”, rainfall is not equal in volume between locations and would vary wildly. Why was this used rather than just the volume?

Page 8 Line 33-47: Drainage density is noted as an important predictor of mosquito-borne breeding. I am unsure if this holds up with the YF sylvatic vector species *Haemagogus* or *Sabethes* mosquitoes which are primarily tree-hole breeding mosquitoes, and as such the drainage does not affect their larval habitats

(https://www.researchgate.net/publication/278666404_Natural_Breeding_Sites_for_Haemagogus_Mosquitoes_Diptera_Culicidae_in_Brazil).

Page 9 Line 41-51: Without absence (or pseudo-absence as the authors have noted it is hard to know where YF is truly absent or just not noticed) the modelling approach is going to be unable to effectively distinguish between areas at risk or not. An approach to get round the issue of whether or not there was actually any YF transmission ongoing is to select pseudoabsence locations based on the presence of other arboviruses (e.g. dengue, zika, chikungunya) and the absence of YF. If there are other arboviruses picked up by the surveillance system, then you can be somewhat more secure in the true absence of YF. These pseudoabsence samples should be of the same size as the presence points.

Methods covariates: There is a general dearth of covariates, I would have expected at least temperature and vegetation (previously found as important, examples <https://journals.plos.org/plosntds/article?id=10.1371/journal.pntd.0005897>, <https://www.nature.com/articles/s41467-021-23926-y>) and measures of seasonality to be included.

Results

Page 12 Line 45-51: I don't know what this “model fit score” is. AUC for each of the model predictions of the categories should be included as this is widely used and generally interpretable. It is currently not possible to see if these models perform any better than flipping a coin.

Page 13 Table 3: The confidence intervals of the parameter estimates should be included. From a quick look at the parameters, it seems that almost no parameters were significant. This should be made clear which are significant.

Page 13 Line 35-50: If the parameters are not significant, then we can't associate them with any outcome. It is not possible to derive the conclusions here if the parameters are not significant (which would be highlighted if the confidence intervals were included).

Page 14 Line 15-24 and Figure 3: The results here are due to the previously mentioned issue of not including absence/pseudo-absence points. The model is unable to appropriately distinguish between areas of risk. I am not convinced by these results, but am unable to back this up with a metric (AUC) because it is not included.

Discussion

Page 18 Line 39-41: “Instead, this model is beneficial for informing ongoing public health practice under current conditions.”. Given how poorly the model performed in 2000-2016 vs 2017, and how the cases and distribution have further changed since, I do not believe this would hold true if compared to current trends.

Page 20 Line 13-16: “Early reports from the 2018-2019 season of YF (defined as December through May) indicated that this season is likely to see fewer cases compared to the 2016-2017 and 2017”. We are now at the end of 2021, this should be updated to reflect on what has happened over the past few years.

Decision letter (RSOS-211305.R0)

Dear Dr Servadio

The Editors assigned to your paper RSOS-211305 "Environmental determinants predicting population vulnerability to high Yellow Fever incidence" have made a decision based on their reading of the paper and any comments received from reviewers.

Regrettably, in view of the reports received, the manuscript has been rejected in its current form. However, a new manuscript may be submitted which takes into consideration these comments.

We invite you to respond to the comments supplied below and prepare a resubmission of your manuscript. Below the referees' and Editors' comments (where applicable) we provide additional requirements. We provide guidance below to help you prepare your revision.

Please note that resubmitting your manuscript does not guarantee eventual acceptance, and we do not generally allow multiple rounds of revision and resubmission, so we urge you to make every effort to fully address all of the comments at this stage. If deemed necessary by the Editors, your manuscript will be sent back to one or more of the original reviewers for assessment. If the original reviewers are not available, we may invite new reviewers.

Please resubmit your revised manuscript and required files (see below) no later than 10-May-2022. Note: the ScholarOne system will 'lock' if resubmission is attempted on or after this deadline. If you do not think you will be able to meet this deadline, please contact the editorial office immediately.

Please note article processing charges apply to papers accepted for publication in Royal Society Open Science (<https://royalsocietypublishing.org/rsos/charges>). Charges will also apply to papers transferred to the journal from other Royal Society Publishing journals, as well as papers submitted as part of our collaboration with the Royal Society of Chemistry (<https://royalsocietypublishing.org/rsos/chemistry>). Fee waivers are available but must be requested when you submit your manuscript (<https://royalsocietypublishing.org/rsos/waivers>).

Thank you for submitting your manuscript to Royal Society Open Science and we look forward to receiving your resubmission. If you have any questions at all, please do not hesitate to get in touch.

on behalf of Dr Krijn Paaijmans (Associate Editor) and Pete Smith (Subject Editor)
openscience@royalsociety.org

Associate Editor Comments to Author (Dr Krijn Paaijmans):

Comments to the Author:

Both reviewers strongly suggest 1) an expansion of covariates considered (factors such as vaccination coverage, temperature, seasonality, and landcover) and 2) to consider the differences between sylvatic and urban transmission of YF in Brazil.

Reviewer 2 points to numerous papers published on environmental/socio-economic suitability of South America to YF, which need to be included in the analysis/bibliography, and asks to employ pseudo-absences in the modelling process, as false absence data can have negative effects on distribution models.

As the amount of work that needs to be done is extensive, and the new analyses can change the manuscript substantially, I recommend that the authors revise and resubmit their manuscript.

Reviewer comments to Author:

Reviewer: 1

Comments to the Author(s)

This paper describes a modelling study with the aims of predicting vulnerability to Yellow Fever (YF) in Brazilian municipalities, based on environmental factors and existing records of the disease. It presents adequate methodology, it is novel and the conclusions are supported by the results. My personal view, however, is that the overall presentation and discussion of the results are somewhat distant from the local dynamics of YF epidemiology in Brazil. Vaccination plays a major role in preventing YF cases, and a low coverage was probably the main determinant of the 2017/2018 epidemic in Minas Gerais. In addition, it would be interesting to improve the discussion on the differences between sylvatic and urban transmission of YF in Brazil, as the paper is centered in environmental determinants. Brazil is endemic for sylvatic YF, and even with the high numbers of the 2017/2018 epidemic, it was not considered as an urban spillover, especially because of the vector species involved in local transmission. Despite the importance of some environmental predictors to vector dynamics, vector's species names are never mentioned in the text. My comments below are attempts to improve the discussion, and do not invalidate the study's main results. Therefore, I believe that a minor revision of the manuscript should be enough for achieving enough quality for publication.

Specific comments and suggestions to the authors:

Abstract:

L6-9: I don't fully agree with the first sentence. I would say that the reasons behind the lack of reported cases in many Brazilian localities are more related with vaccination coverage than with environmental determinants.

L25-26: "...in western and southeastern municipalities..." Brazil is officially divided in five regions: North, Northeast, Southeast, Mid-West and South. Using their recognized names throughout the text would improve interpretation of results and maps.

Introduction:

This section could be improved with a clear justification of why the study was performed in Brazil, and an introduction of the local epidemiology of YF and its dynamics.

Methods:

Do you have any particular reason for not including temperature in the vulnerability models? It is mentioned in the text that it is important for mosquito and YF dynamics, but not included as a variable.

I believe that, because of the way ecoregions were treated in the model, they should be treated as a study limitation and more explored in the discussion. The reasons for making it a binary variable are not clear in the text. I see two potential problems here: 1) from the map in FigS1, I see that you are using what WWF calls biomes and not ecoregions - indeed they do resemble the distribution of the six Brazilian biomes: Amazon, Cerrado, Atlantic Forest, Caatinga, Pantanal, Pampas (maps can be found at IBGE and MMA – Brazilian Ministry of Environment). 2) merging different biomes into two categories (FigS2) might hinder your ability to distinguish their specific effects in the disease outcome.

If you are willing to re-run the analysis, I would suggest treating the six biomes separately. If a 6-level categorical variable is not adequate for your method, I would suggest breaking them down into 6 separate binary variables, to assess the effect of each biome in YF. That can be achieved by reclassifying the values of one biome to 1 and the remaining to 0, for each of the biome classes.

I do not see that as a mandatory correction - you could otherwise keep your current results and be clearer of their limitations - but it would certainly improve your findings and conclusions.

Nevertheless, a great and relevant work.

Reviewer: 2

Comments to the Author(s)

The authors here address an important question, what environmental determinates predict population vulnerability to high YF incidence. While the paper is very well written, and commendably the code is located on github, there are several methodological issues with the analysis and interpretation.

There is an omission of several covariates (EVI, temperature, seasonality, landcover, etc) that have already shown to be important correlates with YF activity, and inclusion of ones not relevant for sylvatic YF (drainage density). These should be expanded considerably. Additionally, while the paper is on the “environmental determinants” there is no reasoning behind omitting additional covariates which may determine the reporting of YF (socio-economics, access to sylvatic habitats, non-human primate presence). The restriction of the dataset to 447 locations, rather than employing pseudoabsence points or another method has led to what appears to be a substantial degree of overfitting, though as they do not provide an interpretable metric of model performance it is hard to say exactly how much. While the authors accurately note that the absence of evidence is not evidence of absence, restricting the dataset without compensating appears to have led to an uninformative model. This is highlighted in the results, as it appears that most of the covariates used in the models are insignificant, and so large portions of the results and discussion are not appropriate.

In a research area that already has numerous papers published on environmental/socio-economic suitability of South America to YF, these omissions are inappropriate. I would strongly recommend that the authors revisit their covariates used, expand them, and employ pseudo-absences in their modelling process, as well as report an interpretable metric on the model fit, only then will we know if the work undertaken here has expanded the field of knowledge on the subject.

Specific comments

Methods

Page 8 Line 20-25: “precipitation data were aggregated to represent the percentage of days for each municipality each year from 2000 through 2017 that experienced any rainfall”, rainfall is not equal in volume between locations and would vary wildly. Why was this used rather than just the volume?

Page 8 Line 33-47: Drainage density is noted as an important predictor of mosquito-borne breeding. I am unsure if this holds up with the YF sylvatic vector species *Haemagogus* or *Sabethes* mosquitoes which are primarily tree-hole breeding mosquitoes, and as such the drainage does not affect their larval habitats (https://www.researchgate.net/publication/278666404_Natural_Breeding_Sites_for_Haemagogus_Mosquitoes_Diptera_Culicidae_in_Brazil).

Page 9 Line 41-51: Without absence (or pseudo-absence as the authors have noted it is hard to know where YF is truly absent or just not noticed) the modelling approach is going to be unable to effectively distinguish between areas at risk or not. An approach to get round the issue of whether or not there was actually any YF transmission ongoing is to select pseudoabsence locations based on the presence of other arboviruses (e.g. dengue, zika, chikungunya) and the absence of YF. If there are other arboviruses picked up by the surveillance system, then you can be somewhat more secure in the true absence of YF. These pseudoabsence samples should be of the same size as the presence points.

Methods covariates: There is a general dearth of covariates, I would have expected at least temperature and vegetation (previously found as important, examples <https://journals.plos.org/plosntds/article?id=10.1371/journal.pntd.0005897>, <https://www.nature.com/articles/s41467-021-23926-y>) and measures of seasonality to be included.

Results

Page 12 Line 45-51: I don't know what this "model fit score" is. AUC for each of the model predictions of the categories should be included as this is widely used and generally interpretable. It is currently not possible to see if these models perform any better than flipping a coin.

Page 13 Table 3: The confidence intervals of the parameter estimates should be included. From a quick look at the parameters, it seems that almost no parameters were significant. This should be made clear which are significant.

Page 13 Line 35-50: If the parameters are not significant, then we can't associate them with any outcome. It is not possible to derive the conclusions here if the parameters are not significant (which would be highlighted if the confidence intervals were included).

Page 14 Line 15-24 and Figure 3: The results here are due to the previously mentioned issue of not including absence/pseudo-absence points. The model is unable to appropriately distinguish between areas of risk. I am not convinced by these results, but am unable to back this up with a metric (AUC) because it is not included.

Discussion

Page 18 Line 39-41: "Instead, this model is beneficial for informing ongoing public health practice under current conditions." Given how poorly the model performed in 2000-2016 vs 2017, and how the cases and distribution have further changed since, I do not believe this would hold true if compared to current trends.

Page 20 Line 13-16: "Early reports from the 2018-2019 season of YF (defined as December through May) indicated that this season is likely to see fewer cases compared to the 2016-2017 and 2017". We are now at the end of 2021, this should be updated to reflect on what has happened over the past few years.

===PREPARING YOUR MANUSCRIPT===

If you have been asked to revise the written English in your submission as a condition of publication, you must do so, and you are expected to provide evidence that you have received language editing support. The journal would prefer that you use a professional language editing service and provide a certificate of editing, but a signed letter from a colleague who is a fluent speaker of English is acceptable. Note the journal has arranged a number of discounts for authors using professional language editing services (<https://royalsociety.org/journals/authors/benefits/language-editing/>).

===PREPARING YOUR REVISION IN SCHOLARONE===

- If you are requesting a discretionary waiver for the article processing charge, the waiver form must be included at this step.
- If you are providing image files for potential cover images, please upload these at this step, and inform the editorial office you have done so. You must hold the copyright to any image provided.
- A copy of your point-by-point response to referees and Editors. This will expedite the preparation of your proof.

- Ensure that your data access statement meets the requirements at <https://royalsociety.org/journals/authors/author-guidelines/#data>. You should ensure that you cite the dataset in your reference list. If you have deposited data etc in the Dryad repository, please include both the 'For publication' link and 'For review' link at this stage.
- If you are requesting an article processing charge waiver, you must select the relevant waiver option (if requesting a discretionary waiver, the form should have been uploaded at Step 3 'File upload' above).
- If you have uploaded ESM files, please ensure you follow the guidance at <https://royalsociety.org/journals/authors/author-guidelines/#supplementary-material> to include a suitable title and informative caption. An example of appropriate titling and captioning may be found at https://figshare.com/articles/Table_S2_from_Is_there_a_trade-off_between_peak_performance_and_performance_breadth_across_temperatures_for_aerobic_scope_in_teleost_fishes_/3843624.

Author's Response to Decision Letter for (RSOS-211305.R0)

See Appendix A.

Decision letter (RSOS-220086.R0)

Dear Dr Servadio

On behalf of the Editors, we are pleased to inform you that your Manuscript RSOS-220086 "Environmental determinants predicting population vulnerability to high Yellow Fever incidence" has been accepted for publication in Royal Society Open Science subject to minor revision in accordance with the referees' reports. Please find the referees' comments along with any feedback from the Editors below my signature.

We invite you to respond to the comments and revise your manuscript. Below the referees' and Editors' comments (where applicable) we provide additional requirements. Final acceptance of

your manuscript is dependent on these requirements being met. We provide guidance below to help you prepare your revision.

Please submit your revised manuscript and required files (see below) no later than 7 days from today's (ie 28-Jan-2022) date. Note: the ScholarOne system will 'lock' if submission of the revision is attempted 7 or more days after the deadline. If you do not think you will be able to meet this deadline please contact the editorial office immediately.

on behalf of Dr Krijn Paaijmans (Associate Editor) and Pete Smith (Subject Editor)
openscience@royalsociety.org

Associate Editor Comments to Author (Dr Krijn Paaijmans):

Associate Editor

Comments to the Author:

The concerns from the reviewers have all been addressed. Reading the manuscript, I noticed that the authors have not mentioned where they obtained the dengue case data. But that's my only and minor comment.

===PREPARING YOUR MANUSCRIPT===

one version should clearly identify all the changes that have been made (for instance, in coloured highlight, in bold text, or tracked changes);

Please ensure that you include an acknowledgements' section before your reference list/bibliography. This should acknowledge anyone who assisted with your work, but does not

qualify as an author per the guidelines at <https://royalsociety.org/journals/ethics-policies/openness/>.

===PREPARING YOUR REVISION IN SCHOLARONE===

- Ensure that your data access statement meets the requirements at <https://royalsociety.org/journals/authors/author-guidelines/#data>. You should ensure that you cite the dataset in your reference list. If you have deposited data etc in the Dryad repository, please only include the 'For publication' link at this stage. You should remove the 'For review' link.
- If you are requesting an article processing charge waiver, you must select the relevant waiver option (if requesting a discretionary waiver, the form should have been uploaded, see 'File upload' above).
- If you have uploaded any electronic supplementary (ESM) files, please ensure you follow the guidance at <https://royalsociety.org/journals/authors/author-guidelines/#supplementary-material> to include a suitable title and informative caption. An example of appropriate titling and captioning may be found at https://figshare.com/articles/Table_S2_from_Is_there_a_trade-off_between_peak_performance_and_performance_breadth_across_temperatures_for_aerobic_scope_in_teleost_fishes_/3843624.

Author's Response to Decision Letter for (RSOS-220086.R0)

See Appendix B.

Decision letter (RSOS-220086.R1)

Dear Dr Servadio,

I am pleased to inform you that your manuscript entitled "Environmental determinants predicting population vulnerability to high Yellow Fever incidence" is now accepted for publication in Royal Society Open Science.

on behalf of Dr Krijn Paaijmans (Associate Editor) and Pete Smith (Subject Editor)
openscience@royalsociety.org

Appendix A

We thank the reviewers for their time spent reading our manuscript and for their comments. We appreciate the points made, as they have strengthened our overall study. Below are specific responses to each of the specific comments made. Our response can be found in blue, indented text underneath the relevant comment.

Associate Editor Comments to Author (Dr Krijn Paaijmans):

Comments to the Author:

Both reviewers strongly suggest 1) an expansion of covariates considered (factors such as vaccination coverage, temperature, seasonality, and landcover) and 2) to consider the differences between sylvatic and urban transmission of YF in Brazil.

Reviewer 2 points to numerous papers published on environmental/socio-economic suitability of South America to YF, which need to be included in the analysis/bibliography, and asks to employ pseudo-absences in the modelling process, as false absence data can have negative effects on distribution models.

As the amount of work that needs to be done is extensive, and the new analyses can change the manuscript substantially, I recommend that the authors revise and resubmit their manuscript.

We appreciate the consideration and ability to revise our manuscript.

Reviewer comments to Author:

Reviewer: 1

Comments to the Author(s)

This paper describes a modelling study with the aims of predicting vulnerability to Yellow Fever (YF) in Brazilian municipalities, based on environmental factors and existing records of the disease. It presents adequate methodology, it is novel and the conclusions are supported by the results. My personal view, however, is that the overall presentation and discussion of the results are somewhat distant from the local dynamics of YF epidemiology in Brazil. Vaccination plays a major role in preventing YF cases, and a low coverage was probably the main determinant of the 2017/2018 epidemic in Minas Gerais. In addition, it would be interesting to improve the discussion on the differences between sylvatic and urban transmission of YF in Brazil, as the paper is centered in environmental determinants. Brazil is endemic for sylvatic YF, and even with the high numbers of the 2017/2018 epidemic, it was not considered as an urban spillover,

especially because of the vector species involved in local transmission. Despite the importance of some environmental predictors to vector dynamics, vector's species names are never mentioned in the text. My comments below are attempts to improve the discussion, and do not invalidate the study's main results. Therefore, I believe that a minor revision of the manuscript should be enough for achieving enough quality for publication.

We appreciate this perspective on our study and its interpretation. We have added text throughout that mentions current vaccine coverage as an important factor in YF spread. We have also clarified that current preventative measures may exist in areas we considered highly vulnerable and our results support their continuation. Estimated vaccination data as of 2016 was found for a point added to the discussion, but not included in revised analyses because the purpose of the study was to assess vulnerability based on environmental conditions that would motivate new or continued vaccine priorities. Additionally, discussion pertaining to sylvatic and urban transmission has been added throughout the revised manuscript.

Specific comments and suggestions to the authors:

Abstract:

L6-9: I don't fully agree with the first sentence. I would say that the reasons behind the lack of reported cases in many Brazilian localities are more related with vaccination coverage than with environmental determinants.

We agree that environmental conditions are not the sole (or necessarily primary) reason that many municipalities in Brazil did not observe YF cases. The statement has been revised, as that was not our intention. The intent was to say that having environmental conditions indicative of susceptibility to disease motivates achieving or maintaining high vaccination coverage.

Based on this comment, wording was changed throughout to acknowledge that locations that are vulnerable to YF burden based on environmental characteristics may currently have high vaccine coverage. Under such circumstances, our work motivates the continuation of currently high coverage. This change can be seen in The first paragraph of section 4, and current vaccination is discussed in the first paragraph of section 4.2.

L25-26: "...in western and southeastern municipalities..." Brazil is officially divided in five regions: North, Northeast, Southeast, Mid-West and South. Using

their recognized names throughout the text would improve interpretation of results and maps.

The names of the regions have been used throughout the abstract and main text in order to increase clarity and consistency.

Introduction:

This section could be improved with a clear justification of why the study was performed in Brazil, and an introduction of the local epidemiology of YF and its dynamics.

The second paragraph of the introduction was expanded to briefly describe YF epidemiology in Brazil and then motivate its use as the study site.

Methods:

Do you have any particular reason for not including temperature in the vulnerability models? It is mentioned in the text that it is important for mosquito and YF dynamics, but not included as a variable.

Temperature has now been included in the analysis in the form of average annual temperature. We were initially hesitant to include a variable that fluctuates as temperature does, but it proved to be an important predictor overall.

I believe that, because of the way ecoregions were treated in the model, they should be treated as a study limitation and more explored in the discussion. The reasons for making it a binary variable are not clear in the text. I see two potential problems here: 1) from the map in FigS1, I see that you are using what WWF calls biomes and not ecoregions - indeed they do resemble the distribution of the six Brazilian biomes: Amazon, Cerrado, Atlantic Forest, Caatinga, Pantanal, Pampas (maps can be found at IBGE and MMA – Brazilian Ministry of Environment). 2) merging different biomes into two categories (FigS2) might hinder your ability to distinguish their specific effects in the disease outcome.

If you are willing to re-run the analysis, I would suggest treating the six biomes separately. If a 6-level categorical variable is not adequate for your method, I would suggest breaking them down into 6 separate binary variables, to assess the effect of each biome in YF. That can be achieved by reclassifying the values of one biome to 1 and the remaining to 0, for each of the biome classes.

The biomes have been expanded into three binary variables. Three of the six biomes were rare in Brazil, and some of these were not present in the municipalities used in model fitting. For this reason, these three were collapsed with the least frequent of the remaining three categories. The wording was also changed throughout the text to refer to these as biomes rather than ecoregions.

I do not see that as a mandatory correction - you could otherwise keep your current results and be clearer of their limitations – but it would certainly improve your findings and conclusions.

Nevertheless, a great and relevant work.

Reviewer: 2

Comments to the Author(s)

The authors here address an important question, what environmental determinates predict population vulnerability to high YF incidence. While the paper is very well written, and commendably the code is located on github, there are several methodological issues with the analysis and interpretation.

There is an omission of several covariates (EVI, temperature, seasonality, landcover, etc) that have already shown to be important correlates with YF activity, and inclusion of ones not relevant for sylvatic YF (drainage density).

These should be expanded considerably. Additionally, while the paper is on the “environmental determinants” there is no reasoning behind omitting additional covariates which may determine the reporting of YF (socio-economics, access to sylvatic habitats, non-human primate presence). The restriction of the dataset to 447 locations, rather than employing pseudoabsence points or another method has led to what appears to be a substantial degree of overfitting, though as they do not provide an interpretable metric of model performance it is hard to say exactly how much. While the authors accurately note that the absence of evidence is not evidence of absence, restricting the dataset without compensating appears to have led to an uninformative model. This is highlighted in the results, as it appears that most of the covariates used in the models are insignificant, and so large portions of the results and discussion are not appropriate.

In a research area that already has numerous papers published on environmental/socio-economic suitability of South America to YF, these omissions are inappropriate. I would strongly recommend that the authors revisit their covariates used, expand them, and employ pseudo-absences in their modelling process, as well as report an interpretable metric on the model fit, only then will we know if the work undertaken here has expanded the field of knowledge on the subject.

We appreciate this perspective and the attention to methods used. Our analyses were changed to reflect these comments. The major changes in analytic methods were the use of pseudo-absences (defined by locations observing dengue but not YF), addition of temperature and vegetation as candidate predictors, expanding ecoregions/biomes into three categories (as suggested by Reviewer 1), and using AUC to determine model fit rather than our own metric.

Specific comments

Methods

Page 8 Line 20-25: “precipitation data were aggregated to represent the percentage of days for each municipality each year from 2000 through 2017 that experienced any rainfall”, rainfall is not equal in volume between locations and would vary wildly. Why was this used rather than just the volume?

The frequency of rainfall was used in order to represent number of opportunities for standing water to be found. This is particularly pertinent to urban transmission, as *Aedes* mosquitoes can lay eggs in small quantities of standing water. While the volume of water is also an important factor, knowing the total rainfall does not distinguish whether there were fewer days with higher rainfall or more days with lower rainfall.

Page 8 Line 33-47: Drainage density is noted as an important predictor of mosquito-borne breeding. I am unsure if this holds up with the YF sylvatic vector species *Haemagogus* or *Sabethes* mosquitoes which are primarily tree-hole breeding mosquitoes, and as such the drainage does not affect their larval habitats

(https://www.researchgate.net/publication/278666404_Natural_Breeding_Sites_of_Haemagogus_Mosquitoes_Diptera_Culicidae_in_Brazil).

While *Haemagogus* and *Sabethes* mosquitoes may be less sensitive to water drainage, this factor was potentially pertinent to *Aedes* mosquitoes. However, with the candidate covariates added to the analyses and changed model fit criteria, this variable was not retained in the best-fitting models for either time period. Additionally, to better address the fact that consideration that different mosquito genera behave differently, text was added to contextualize our methods and results for urban and sylvatic

mosquitoes. This can be found in the fourth paragraph of section 1 and throughout section 2.2.2,

Page 9 Line 41-51: Without absence (or pseudo-absence as the authors have noted it is hard to know where YF is truly absent or just not noticed) the modelling approach is going to be unable to effectively distinguish between areas at risk or not. An approach to get round the issue of whether or not there was actually any YF transmission ongoing is to select pseudoabsence locations based on the presence of other arboviruses (e.g. dengue, zika, chikungunya) and the absence of YF. If there are other arboviruses picked up by the surveillance system, then you can be somewhat more secure in the true absence of YF. These pseudoabsence samples should be of the same size as the presence points.

This has been added. An additional 466 municipalities where dengue was seen in 2006, but YF was not seen, were added to analyses.

Methods covariates: There is a general dearth of covariates, I would have expected at least temperature and vegetation (previously found as important, examples <https://journals.plos.org/plosntds/article?id=10.1371/journal.pntd.0005897>, <https://www.nature.com/articles/s41467-021-23926-y>) and measures of seasonality to be included.

This comment is consistent with Reviewer 1 and previous literature. Temperature and vegetation have been included in updated analyses. A measure of seasonality was not included because data were analyzed by year, and therefore each data observation encompasses a season of Yellow Fever.

Results

Page 12 Line 45-51: I don't know what this "model fit score" is. AUC for each of the model predictions of the categories should be included as this is widely used and generally interpretable. It is currently not possible to see if these models perform any better than flipping a coin.

Model fit was repeated using AUC of an ROC curve instead of the previous method.

Page 13 Table 3: The confidence intervals of the parameter estimates should be included. From a quick look at the parameters, it seems that almost no

parameters were significant. This should be made clear which are significant.

Confidence intervals have been added in place of standard errors in Table 3.

Page 13 Line 35-50: If the parameters are not significant, then we can't associate them with any outcome. It is not possible to derive the conclusions here if the parameters are not significant (which would be highlighted if the confidence intervals were included).

The confidence intervals make it more apparent that, in the updated analyses, few predictors were significant in the 2000-2016 model, and most were significant in the 2017 model. It is noted in the text that many of the predictors with low magnitude were nonsignificant. It is, however, important to retain them in the model because their inclusion still improves predictive ability, seen through the AUC. In the context of these models, nonsignificance reflects a low magnitude association with low precision more than total irrelevance because the model aim was to maximize prediction. (Non)significance of predictors is mentioned in the third and fourth paragraphs of section 3.1, and this has been added to the fourth paragraph of section 4.2.

Page 14 Line 15-24 and Figure 3: The results here are due to the previously mentioned issue of not including absence/pseudo-absence points. The model is unable to appropriately distinguish between areas of risk. I am not convinced by these results, but am unable to back this up with a metric (AUC) because it is not included.

Our updated results were selected through AUC and show that the model can discriminate well across our categories. It has been noted in the discussion that, particularly for the 2000-2016 model, the model may not be detecting influence from environmental factors, only the spatial relationship as municipalities with cases tend to be near each other.

Discussion

Page 18 Line 39-41: "Instead, this model is beneficial for informing ongoing public health practice under current conditions.". Given how poorly the model performed in 2000-2016 vs 2017, and how the cases and distribution have further changed since, I do not believe this would hold true if compared to current

trends.

An important limitation of our analyses is that they are not able to detect major shifts in disease dynamics, as evidenced by the inability of the 2000-2016 model to predict the 2016-2018 outbreak. This has been reworded to better reflect the actual interpretation and implication, stating that the models would need updating to inform today's preparedness strategy. This has been added to the second paragraph of section 4.1.

Page 20 Line 13-16: "Early reports from the 2018-2019 season of YF (defined as December through May) indicated that this season is likely to see fewer cases compared to the 2016-2017 and 2017". We are now at the end of 2021, this should be updated to reflect on what has happened over the past few years.

More recent information was acquired, and the 2018-2019 and 2019-2020 seasons were used for comparison. The 2020-2021 season was not included in case the ongoing COVID-19 pandemic led to changes in human activity that affected YF burden in Brazil. This has been added to the second paragraph of section 4.1.

Appendix B

Associate Editor Comments to Author (Dr Krijn Paaijmans):

Associate Editor

Comments to the Author:

The concerns from the reviewers have all been addressed. Reading the manuscript, I noticed that the authors have not mentioned where they obtained the dengue case data. But that's my only and minor comment.

JLS: We appreciate the second review of our manuscript and are pleased that our revised version was found to adequately address previous comments.

Thank you for catching this oversight. The data source for the dengue case data, a publicly available dataset from Brazil's Ministry of Health, has been stated in the text with an appropriate citation.